# *Tradescantia*-Based Test Systems Can Be Used for the Evaluation of the Toxic Potential of Harmful Algal Blooms

Alla Khosrovyan [1,*], Rima Avalyan [2], Anahit Atoyants [2], Evelina Aghajanyan [2], Lusine Hambaryan [3], Rouben Aroutiounian [2] and Bardukh Gabrielyan [4]

1. National Institute of Chemical Physics and Biophysics, 23 Akadeemia Tee, 12618 Tallinn, Estonia
2. Laboratory of General and Molecular Genetics, RI Biology, Faculty of Biology, Yerevan State University, 8 Charents Str., Yerevan 0025, Armenia; r.avalyan@ysu.am (R.A.); anahit.atoyants@ysu.am (A.A.); evagadjanyan@ysu.am (E.A.); genetik@ysu.am (R.A.)
3. Chair of Ecology and Nature Conservation, Faculty of Biology, Yerevan State University, 8 Charents Str., Yerevan 0025, Armenia; lusinehambaryan@ysu.am
4. Scientific Center of Zoology and Hydroecology, National Academy of Sciences of Armenia, 7 Paruir Sevak Str., Yerevan 0014, Armenia; gabrielb@sci.am
* Correspondence: alla.khosrovyan@kbfi.ee

**Abstract:** Harmful algal blooms (HABs) are overgrowths of toxic strains of algae (diatoms, green) and cyanobacteria (blue-green algae). While occurring naturally, human-induced environmental changes have resulted in more frequent occurrences of such blooms worldwide. Meantime, the ecotoxicological risk of HABs is rarely evaluated by means of standard test methods. For the first time, the genotoxic potential of the HAB event 2020 was assessed using two different *Tradescantia*-based test systems (Trad-SHM and Trad-MN, 24-h exposure). An integrated analysis of biological (algal abundance) and ecotoxicological (testing) data revealed linkages among algal proliferation, changes in *Tradescantia* stamen hairs (mutations and suppressed growth) and chromosomal aberrations during microsporogenesis (appearance of micronuclei) that were likely to be caused by toxic algal groups. Green alga *Botryococcus braunii* and the cyanobacterial species *Anabaena* and *Oscillatoria* could suppress stamen hair growth; Cyanobacteria *Phormidium* and *Aphanothece* sp. could trigger mutations in stamen hairs (appearance of pink and colorless cells); and *Oscillatoria* sp. could be responsible for the occurrence of chromosomal damage. Diatom proliferation in the spring was not related to the genotoxic response in *Tradescantia*. Both tests, the Trad-SHM and Trad-MN, are suitable for the evaluation of the toxic potential of HABs.

**Keywords:** somatic cell; mutation; genotoxicity; stamen hair mutation test; micronuclei test; *Tradescantia* inflorescences





## 1. Introduction

Harmful algal blooms (HABs), the overgrowth of toxic strains of algae (diatoms, green) and cyanobacteria (known as blue-green algae) occur naturally in lakes and slow-moving rivers. HABs pose direct toxicity to fish and humans, create certain environmental risks via the formation of oxygen-depleted water (so-called dead zones), affect ecosystem services and disrupt aquatic recreation. The toxins produced by freshwater diatoms and cyanobacteria enter the human body directly from drinking water, recreational exposure and the consumption of agricultural produce irrigated by contaminated water or seafood from contaminated areas [1,2]. For example, saxitoxin (a neurotoxin known as paralytic shellfish poison) enters organisms by ingesting contaminated seafood and can cause acute toxicity within days [3,4]. The liver, gut and muscle of fish inhabiting the lakes affected by algal blooms were found to contain microcystins—natural toxins produced by *Microsystis* and *Anabaena* [5]. The consumption of contaminated food may result in human and animal intoxication [6]. Massive fish kills, which occurred in aquaculture facilities in the

Arabian Gulf, Canada and Malaysia, were associated with the blooming of dinoflagellate *Cochlodinium* sp. [7–9]. Changes in the zooplankton assemblages were also related to summer and winter HAB events in a Mediterranean lagoon in Greece, causing a sharp decline of all taxa, including the disappearance of some key copepod species [10]. Although grazing on toxic algae does not always cause adverse effects on copepods, ingestion of toxic algae during HABs may diminish their survival, egg production or egg hatching [11]. Although the factors causing the blooms are not yet fully understood [12], it is commonly acknowledged that human-induced environmental changes have resulted in more frequent occurrences of such blooms (https://www.noaa.gov/what-is-harmful-algal-bloom, accessed on 4 June 2023).

Lake Sevan (Armenia), a mountain lake situated at 1900.5 m asl, is an oligotrophic water body receiving discharge from 28 rivers (Figure 1). The lake comprises two morphologically different parts: a northern deep part (with an average/maximum depth of 35.82/83.62 m), the Small Sevan (SS), and a southern shallow part (with an average/maximum depth of 22.62/43.62 m), the Big Sevan (BS) (data on the location and depth are for 2022, provided by the Institute of Geology of Armenia). Anthropogenic activity, such as lowering the water level, habitat degradation, land use in the catchment area, and overfishing, has severely affected the lake's ecosystem [13]. In particular, land use changes were followed by an increased nutrient supply from the catchment area, especially from the rivers. In the 1980s, 57% of nitrogen and 45% of phosphorus entered the lake from the river flow, while only 16% of nitrogen and 27% of phosphorus entered from the diffuse sources [14]. An intensive input of nutrients from the catchment area resulted in an increase in the total average annual algal biomass: from 0.2–0.5 g m$^{-3}$ (1937–1962) to 2.0–3.0 g m$^{-3}$ (1976–1983) [13,14]. The first cyanobloom in the lake (*Anabaena flos-aquae* and *Aphanizomenon flos-aquae*) occurred in 1964 [15].

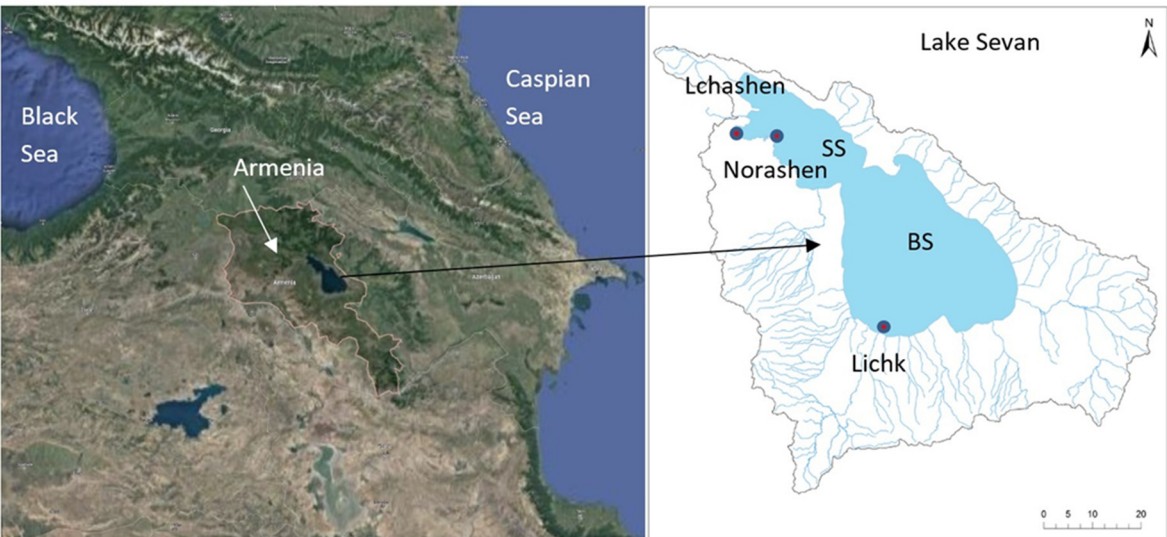

**Figure 1.** Location of sampling sites in the bays of Lake Sevan (Armenia). SS (Small Sevan)—northern part of the lake; BS (Big Sevan)—southern part. Bays: Lchashen, Norashen and Lichk.

Over time, the diversity of the phytoplankton community decreased and an unpredictable succession of species became frequent [16]. From 2018 to 2020, the mean annual phytoplankton biomass doubled in the littoral (from 8 g m$^{-3}$ to 16 g m$^{-3}$). The share of cyanobacterial species increased in the phytoplankton community, and HAB events were recorded in 2018 and 2020 (Figure 2).

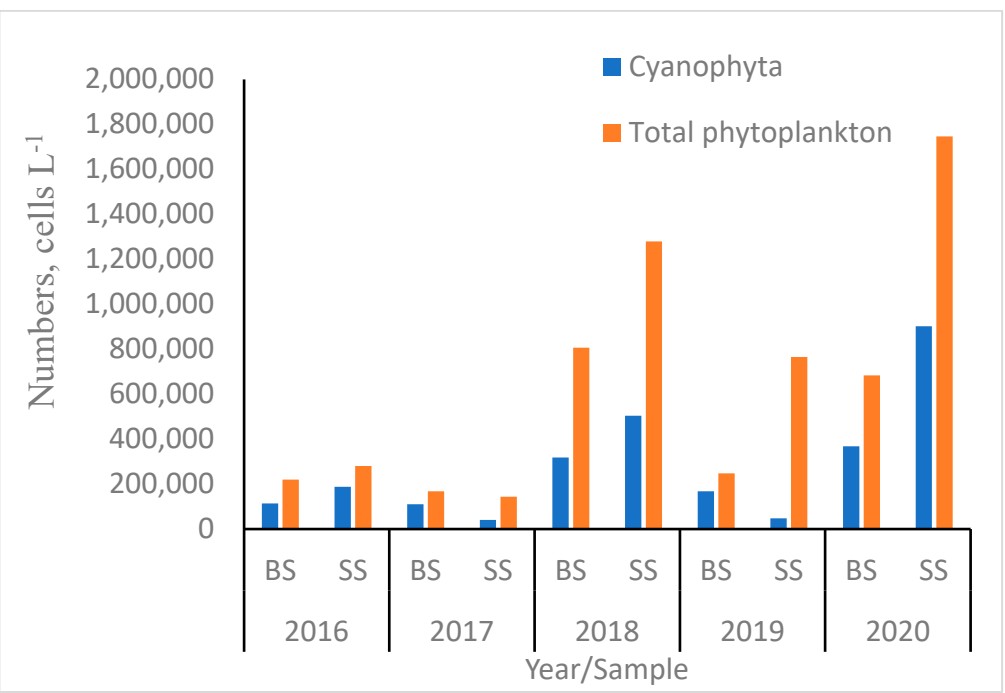

**Figure 2.** Mean total algal and cyanobacterial numbers in the month of July in different parts of Lake Sevan (Armenia) by year. SS—northern part of the lake; BS—southern part.

HAB events may be also observed in rivers. It is commonly known that the blooms occur in slow-moving waters. Yet, the Ohio River cyanoHAB event of 2015 resulted in green water flowing for hundreds of kilometers, starting 135 km from the river's origin and reaching its confluence with the Mississippi River in 30 days ([17]; https://www.epa.gov/sciencematters/better-way-application-risk-characterization-habs-ohio-river, accessed on 4 June 2023). Such a phenomenon may suggest that even moving rivers can be susceptible to an algal bloom or its consequences, such as oxygen reduction [18,19] or cyanotoxin occurrence [20,21].

Considering the predicted climate warming, HABs are likely to become more frequent worldwide, and hence, the assessment of risk for resident aquatic communities is of great importance. However, a search in the Web-of-Science database, with the keywords "HAB" and "toxic*" resulted in 906 hits, which were mainly devoted to monitoring, mitigating, controlling and predicting harmful algal blooms. According to this dataset, few studies assessed the toxicity of HABs. For example, the macrophyte *Ceratophyllum submersum* showed physiological stress under HAB events [22]. The swimming behavior of spermatozoa and the egg viability of pearl oysters, *Pinctada fucata martensii*, were affected by their exposure to toxic dinoflagellates and raphidophytes [23]. Microtox (R) testing detected toxicity in the samples collected during cyanobacteria, dinoflagellate and haptophyte blooms [24]. However, no studies used the *Tradescantia* test for the evaluation of HABs' potential toxicity, although the *Tradescantia*-based test systems are known for their high sensitivity to even low concentrations of contaminants and wide applicability for a variety of risk assessment purposes (air, water, soils) [25–28]. For example, the Trad-SHM and Trad-MN test systems with *Tradescantia* clone 02 showed toxic effects, even when most of the measured toxicants in a lake were below the WHO standards for drinking water quality [29], suggesting high sensitivity to low contaminant concentrations [25] or high performance in the detection of the potential toxicity of a contaminant mix. *T. pallida* was employed in a 2 h exposure to diesel and biodiesel exhaust emissions, demonstrating higher genotoxicity, consistent with higher concentrations of particulate matter, carbonyls and PAH content [28]. In this work, for the first time, the genotoxic potential of a HAB event was assessed using two different *Tradescantia*-based test systems on HAB in Lake Sevan (Armenia) in July 2020. Our

working hypothesis was that the toxic potential of HABs could be determined by standard *Tradescantia*-based test systems.

## 2. Materials and Methods

Study Sites and Sampling

Three bays of Lake Sevan (Armenia), situated on the eastern side of the lake, were investigated. Bays represent biodiversity-rich environments due to their partial surroundings by land, specific hydrological conditions, low depth, and increased heating of the water column. Lichk Bay, into which two rivers empty, is located in the southern part of the lake (in BS). Norashen and Lchashen bays, located in the northern part of the lake (in SS), are not fed by rivers. All three bays receive inputs from various anthropogenic sources (domestic wastewater, agricultural runoff, industrial activity). The water samples were collected in 2020 during three different seasons—spring (25th of May), summer (7th of July) and autumn (5th of October). In July, a HAB event occurred in Lichk Bay.

Chemical analysis

The concentration of algal nutrients (ammonium and phosphate) and other ions (F, Cl, $HCO_3^-$, $SO_4^{2-}$) was determined by using standard methods by the Laboratory of Hydrogeochemistry of the Institute of Geological Sciences (Armenia). The concentration of metals (K, Na, Ca, Mg, total Fe, Cu, Zn, Pb and Cd) in water was measured on an ICP-MS inductively coupled plasma mass spectrometer by the Hydrometeorology and Monitoring Center under the Ministry of Environment of Armenia.

Hydrometeorological data

Data on the mean air temperature in the study area at different seasons were obtained from the Armenia State Hydrometeorological and Monitoring Service (Figure S1).

Algae and cyanobacteria

A 1 L sample per site (0.5 m depth) was preserved with a 40% formaldehyde solution and stored for 10 days in darkness. After concentrating the solution to a 10 mL volume by standard methods [30,31], the phytoplankton was identified to the genera or species level using the key determinants [32–36]. The numbers of phytoplankton by species were counted using the Nageotte counting chamber (0.1 mL) [37,38].

Bioassays

Two standard tests with the plant *Tradescantia* (clone 02), described by [39,40], were used for assessing the potential toxicity of the HAB event that occurred in 2020. Briefly, the first test is a stamen hair mutation test (Trad-SHM), which allows for the evaluation of the genotoxicity potential of contaminants. It scores the changes in the heterozygous dominant blue color of the stamen hair cells to pink color (a mutation of the dominant gene) and the appearance of stunted hairs (hair containing less than 12 cells). The second test is a micronuclei test (Trad-MN), which allows the evaluation of the clastogenic potential of a contaminant(s). It scores the micronuclei (MN) frequency in pollen mother cells and detects disturbances in the process of microsporogenesis. For both tests, three to seven young *Tradescantia* inflorescences were submerged in the water samples for 24 h at room temperature. In the Trad-SHM test, after a recovery period of 7 days, opened flowers were daily analyzed for the appearance of pink cells (PC), colorless cells (CC) and stunted hairs (SH). Inflorescences were collected and analyzed over 21 days. The number of mutations was calculated per 1000 hairs. An amount of 10,000 to 20,000 stamen hairs were analyzed per sample.

In the Trad-MN test, inflorescences were fixed in Carnoy's solution (3:1 ethanol-glacial acetic acid) for 24 h and stained with 0.5% acetocarmine. The number of MN and the number of tetrads with MN were calculated per 100 tetrads. A total of 3000 tetrads were analyzed per sample. Both tests were performed in triplicate. As a negative control,

dechlorinated tap water was used. As a positive control, the 10 mM chromium (VI) oxide ($CrO_3$) solution was used.

Statistical analysis

To determine the statistical differences in the genotoxicity indicators among treatments and between treatments and the control, one-way ANOVA with Tukey's post hoc test was used (the significance level was set to 5%). A Pearson correlation analysis between the related responses (MN and Tetrads with MN) was conducted after the normality check of their distributions. An integrated analysis of biological and ecotoxicological data was conducted using principal component analysis (PCA) to reveal linkages among the nutrient concentration, algal bloom (algal and cyanobacterial numbers) and genotoxicity indicators (PC, CC, SH, MN). For this, only those algal and cyanobacterial species that were present for most of the seasons at each site were chosen to avoid the missing data in datasets. PCA was run for the total algal/cyanobacterial numbers and for the individual species separately. Biplots of factor scores and factor loadings ($\geq$0.4) were interpreted for interpreting the linkages between variables. The statistical analysis was performed in R (version 4.1.3 2022) using the corrr, ggcorrplot, FactoMineR and factoextra packages.

## 3. Results

### 3.1. Chemical Analysis

The results of the chemical analyses of the water samples are shown in Table 1. The concentrations of all measured metals (K, Na, Ca, Mg, total Fe, Cu, Zn, Pb, Cd) and other (Cl, $HCO_3^-$, $SO_4^{2-}$) ions were far below the "average" water quality index, mostly conforming to "good" and even "excellent" water quality criterion, according to the national surface water quality criteria adopted by the Ministry of Environment of Armenia (http://env.am/en/environment/environmental-monitoring, accessed on 4 June 2023). The concentration of fluoride ion ($F^-$) was far below the WHO's drinking water quality guidelines of 1.5 mg $L^{-1}$ [41].

**Table 1.** Concentrations of select elements present in the water samples of Lichk, Norashen and Lchashen bays of Lake Sevan (Armenia) in 2020 by season.

| | $K^-$ | $Ca^{2+}$ | $Na^+$ | $Mg^{2+}$ | $F^-$ | $Cl^-$ | $SO_4^{2-}$ | $HCO_3^-$ | Fe Total | $Cu^{2+}$ | $Zn^{2+}$ | $Pb^{2+}$ | $Cd^{2+}$ |
|---|---|---|---|---|---|---|---|---|---|---|---|---|---|
| | | | | | | | mg $L^{-1}$ | | | | | | |
| **Spring** | | | | | | | | | | | | | |
| Lichk | 7.8 | 31 | 43 | 21 | 0.4 | 44 | 19 | 220 | 0.26 | <dL | <dL | <dL | <dL |
| Norashen | 15 | 27 | 55 | 46 | 0.65 | 53 | 39 | 366 | 0.04 | <dL | <dL | <dL | <dL |
| Lchashen | 15 | 27 | 65 | 45 | 0.68 | 52 | 38 | 366 | 0.04 | <dL | <dL | <dL | <dL |
| **Summer** | | | | | | | | | | | | | |
| Lichk | 20.2 | 27 | 60 | 40 | 0.65 | 56 | 34 | 337 | 0.07 | <dL | <dL | <dL | <dL |
| Norashen | 28.27 | 28 | 65 | 45 | 0.65 | 67 | 35 | 366 | 0 | <dL | <dL | <dL | <dL |
| Lchashen | 34.09 | 27 | 63 | 45 | 0.68 | 69 | 33 | 366 | 0.04 | <dL | <dL | <dL | <dL |
| **Autumn** | | | | | | | | | | | | | |
| Lichk | 12.35 | 27 | 54 | 42 | <dL | <dL | <dL | <dL | 0.02 | <dL | <dL | <dL | <dL |
| Norashen | 13.5 | 27 | 61 | 45 | <dL | <dL | <dL | <dL | 0.04 | <dL | <dL | <dL | <dL |
| Lchashen | 13.5 | 27 | 60 | 46 | <dL | <dL | <dL | <dL | 0.02 | <dL | <dL | <dL | <dL |
| "average" water quality * | 4 × B | 200 | 4 × B | 100 | - | 150 | 150 | - | 0.5 | 0.05 | 0.2 | 0.025 | B + 0.002 |

**Table 1.** *Cont*.

| | K⁻ | Ca²⁺ | Na⁺ | Mg²⁺ | F⁻ | Cl⁻ | SO₄²⁻ | HCO₃⁻ | Fe Total | Cu²⁺ | Zn²⁺ | Pb²⁺ | Cd²⁺ |
|---|---|---|---|---|---|---|---|---|---|---|---|---|---|
| "good" quality * | $2 \times B$ | 100 | $2 \times B$ | 50 | - | $2 \times B$ | $2 \times B$ | - | $2 \times B$ | $B + 0.02$ | | | |
| "excellent" quality * | B | B | B | B | - | B | B | - | B | B | | | |

Notes: * Armenia surface water quality guidelines. B: background value of a given element in a local environment; - not available; dL detection limit: 0.002 mg L⁻¹.

The concentrations of select biogenic elements (algal nutrients) in the water at different seasons are shown in Figure 3.

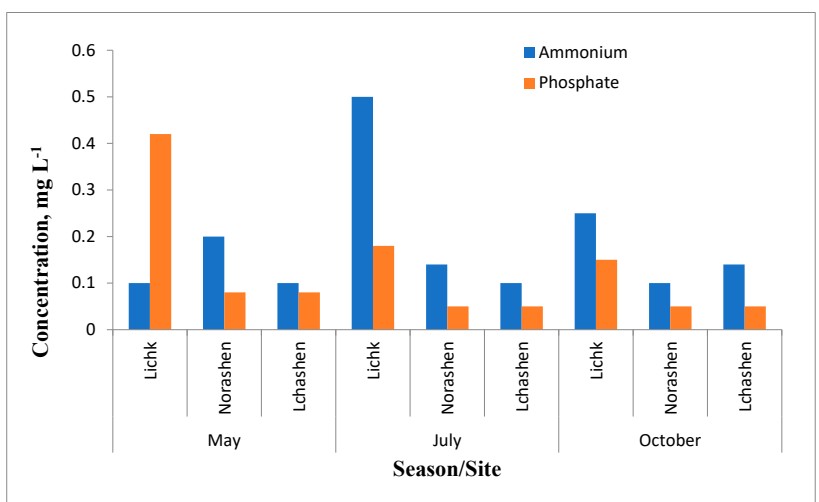

**Figure 3.** Ammonium and phosphate concentrations in water at Lichk, Norashen and Lchashen bays of Lake Sevan (Armenia) in 2020 by season.

*3.2. Biological and Ecotoxicological Analyses*

The abundance of the main phytoplankton phyla at different seasons is shown in Figure 4.

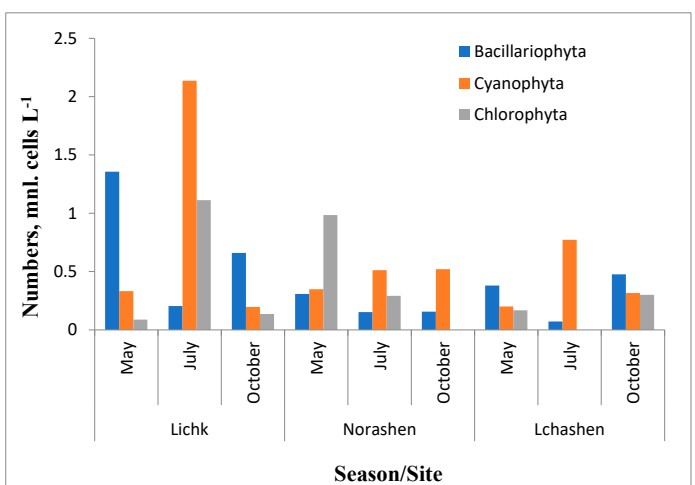

**Figure 4.** Total phytoplankton numbers by main algal/cyanobacterial phyla at Lichk, Norashen and Lchashen bays of Lake Sevan (Armenia) in 2020 by season.

The abundance of various cyanobacterial and green algal genera by season is shown in Figure 5.

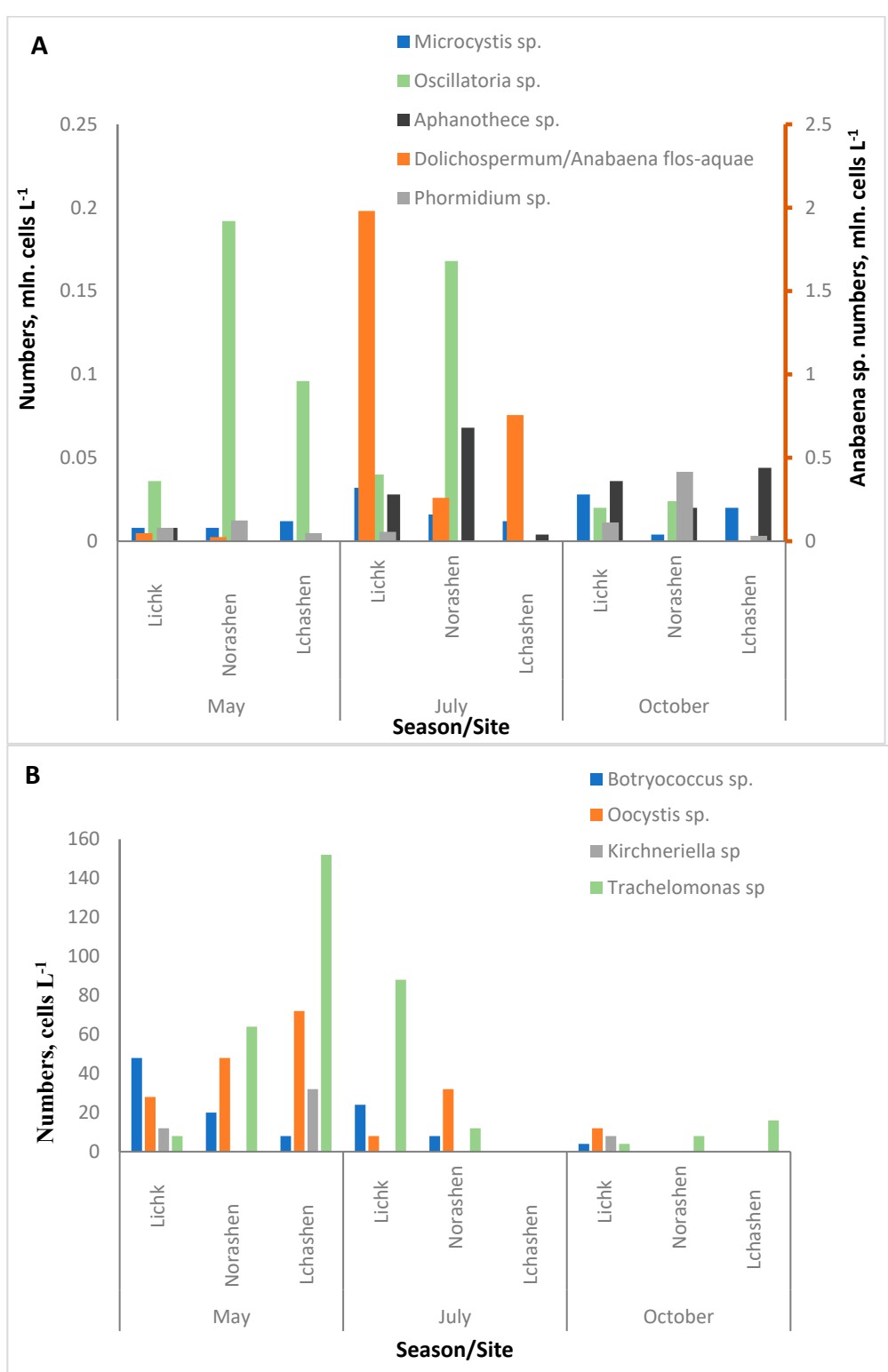

**Figure 5.** Total numbers of Cyanophyta (**A**) and Chlorophyta (**B**) genera in Lichk, Norashen and Lchashen bays of Lake Sevan (Armenia) in 2020 by season.

The frequencies of occurrence of the study genotoxicity indicators are shown in Figure 6.

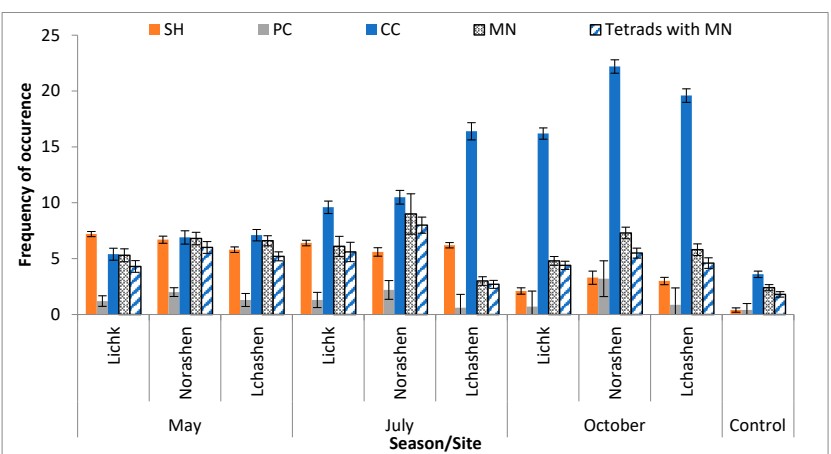

**Figure 6.** Genotoxicity responses in *Tradescantia* (clone 02) exposed to water samples from Lichk, Norashen and Lchashen bays of Lake Sevan (Armenia) in 2020 by season (mean $\pm$ SD). Responses: SH—stunted stamen hair; PC—pink cells in the stamen hair; CC—colorless cells in the stamen hair; micronuclei (MN), tetrads with MN. Control: municipal tap water. All responses are significantly different compared to the control ($p < 0.05$), except for those denoted by "I".

### 3.3. Integrated Analysis of Biological and Ecotoxicological Data

For the different datasets, three principal components were extracted. In all PCA runs, the extracted components together explained more than 83% of the total variance, while each principal component explained nearly 30% of the total variance in a dataset. The results of the PCA analysis with total numbers of each phylum, Cyanophyta (blue-green), Chlorophyta (green) and Bacillariophyta (diatom), are shown in Figure 7. Due to a high positive correlation between MN and Tetrads with MN (R = 0.96, Pearson correlation analysis, significance level 0.01), only MN values were used in the PCA.

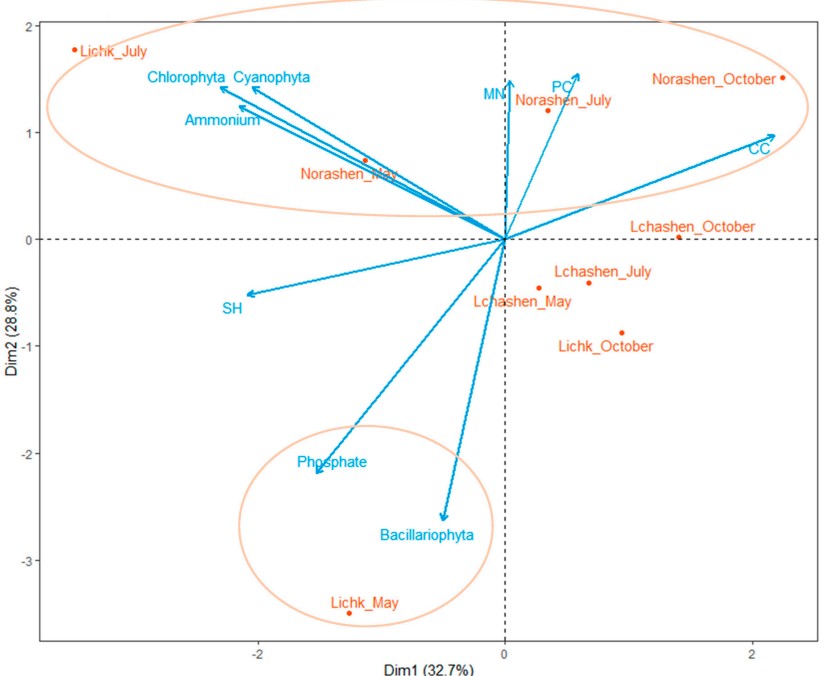

**Figure 7.** Biplot of component loadings and scores for genotoxicity indicators (PC, CC, SH, MN), algal nutrients (ammonium, phosphate), algal/cyanobacterial abundance (total numbers) and sampling sites (Lichk, Norashen and Lchashen bays) in the coordinate system of two principal components. In parentheses: percent of total variance explained by each component.

The relationships between the abundance of individual Cyanophyta and Chlorophyta species and the genotoxicity indicators are shown in Figures 8 and 9. The Bacillariophyta abundance by genera/species is shown in Figure S2.

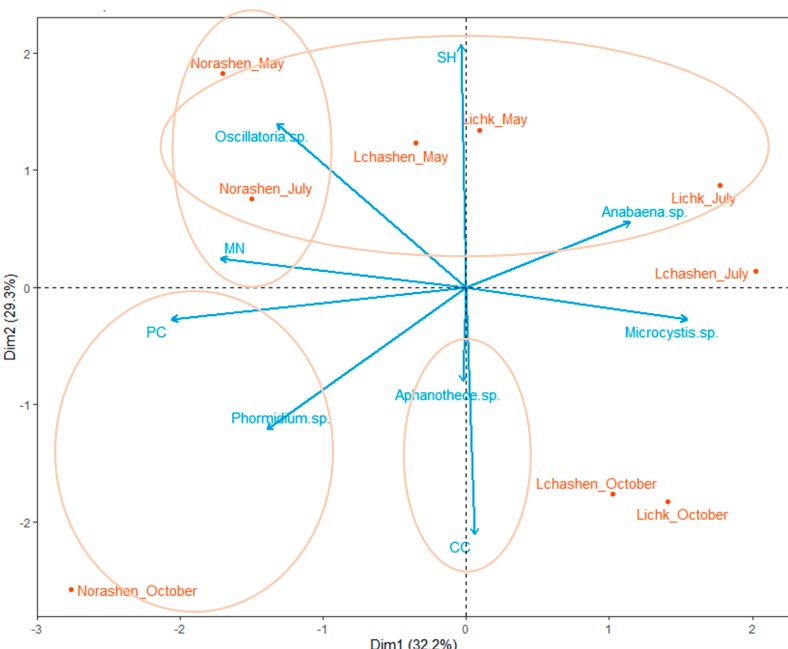

**Figure 8.** Biplot of component loadings and scores for genotoxicity indicators (PC, CC, SH, MN), cyanobacterial species abundance (total numbers) and sampling sites (Lichk, Norashen and Lchashen bays) in the coordinate system of two principal components. In parentheses: percent of total variance explained by each component.

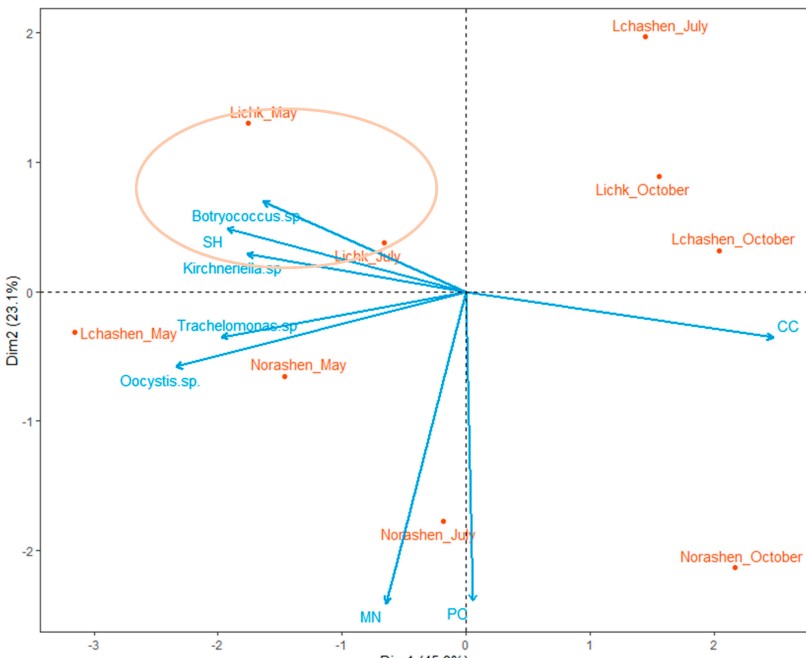

**Figure 9.** Biplot of component loadings and scores for genotoxicity indicators (PC, CC, SH, MN), green algae species abundance (total numbers) and sampling sites (Lichk, Norashen and Lchashen bays) in the coordinate system of two principal components. In parentheses: percent of total variance explained by each component.

## 4. Discussion

The genotoxic potential of the HAB event of 2020 was evaluated by means of the Trad-SHM and Trad-MN tests and the integrated analysis of chemical, biological and eco-toxicological data (PCA). Numerous relationships were revealed among the algal nutrient availability in water, abundance of various phytoplankton taxa and the genotoxicity indicators at various seasons: before (May), during (July) and after (October) the HAB event.

Algal nutrients availability and algal blooming

Appropriate climatic conditions and nutrient availability foster the proliferation of algae in aquatic systems. However, specific ecological requirements of different species (e.g., related to the hydrological regime, light and temperature) and their nutrient acquisition strategies may influence the proliferation of certain groups. PCA grouped the total numbers of Chlorophyta and Cyanophyta with ammonium and phosphate concentrations at Lichk in the summer (Figure 7). At Lichk, the ammonium load was the greatest from the summer to autumn, compared to the other bays (Figure 3), although it corresponded to the "good" water quality (according to the above-mentioned national surface water quality criteria). Different forms of a specific nutrient element do not equally contribute to the growth of phytoplankton [12], however, easily accessible phosphate and ammonium are key elements for photoautotrophs [42]. Ammonium is incorporated into carbon skeletons of photosynthetic organisms, and hence they prefer to uptake it directly from the medium without the energy expense of a reduction in other nitrogen forms to ammonium [43,44].

The phosphate ion concentration in the summer corresponded to a nearly "average" water quality, according to the above-mentioned criteria (Figure 3), indicating phosphorus enrichment in the bay. Hence, the highest abundance of Chlorophyta and Cyanophyta observed in the summer at Lichk Bay was likely caused by the availability of algal nutrients and appropriate environmental conditions. Chlorophyta was shown to be abundant in standing or slow-moving waters enriched with nutrients and when light and temperature are relatively high [45]. Similarly, cyanobacteria also benefit from increased water temperatures, which lead to a reduction in turbulent mixing of the water column and prolonged stratification periods [46] and, hence, stimulate a high cell replication rate [47]. At Lichk, where the nutrient concentrations were the greatest among the sites in the summer, HAB caused by Cyanophyta reached 2,136,000 cells $L^{-1}$ with Dolichospermum/Anabaena flos-aquae amounting to 1,980,000 cells $L^{-1}$ (Figure 5).

PCA also grouped the Bacillariophyta (diatom) numbers with phosphate concentrations at Lichk in the spring (Figure 7). The phosphate concentration exceeded 0.4 mg $L^{-1}$, corresponding to the "unsatisfactory" water quality index. Diatoms proliferated in the spring, reaching 1,356,000 cell $L^{-1}$ (Figure 4). Diatoms, the most abundant species in the phytoplankton [48], often dominate in spring [49]. This is likely stipulated by the peculiar water mixing regime, which increases the availability of algal nutrients. Hydrodynamics play a significant role in triggering diatom blooms, especially of the lotic diatom genera (such as *Cymbella*, *Nitzshia* and *Navicula* [50,51]), which adopted appropriate growth and nutrient uptake strategies. For example, slight turbulence enhances the diffusion rate of nutrients to the cell surface of lotic and/or phosphorus-tolerant *diatoms Cyclotela atomus, Fragilaria crotonensis,* and *Nitzshia palea*, increasing their growth rates [50]. At Lichk, the share of lotic diatoms, including *F. crotonensis, N. palea, N. linearis, N. acicularis, A. ovalis, Navicula radioza, C. ventricose, Caloneis silicula, Cocconeis placentula, Diatoma vulgaris* and *Surirella ovata* in the spring, exceeded 24%.

HAB and genotoxic responses of *Tradescantia*

One of the main concerns about HABs is the exposure to cyanotoxins. A release of toxins from diatom/cyanobacteria in the summer–autumn periods was well demonstrated for various water bodies (e.g., [4,52–54]). In a past HAB event in Lake Sevan (July 2018), the total (cell-bound only) microcystin concentration exceeded 2.4 μg $L^{-1}$ [55]. In 2019, the anatoxin-a concentration in Lake Sevan's water reached ~0.15 μg $L^{-1}$ [56]. While summer is a characteristic period for cyanobacterial blooms [49], toxin production and

release can also occur during diatom blooms from winter to spring [4]. The production of toxins by algae/cyanobacteria and, hence, toxin bioavailability, depends on many factors. For example, phosphorus availability stimulates the growth of microcystin production by *Microcystis aeruginosa*. However, when the phosphorus was limited, the microcystin content increased (even towards producing more toxic forms of microcystin), despite the reduced growth of *M. aeruginosa* [57]. Similarly, at the phosphate concentration of 0.1 mg $L^{-1}$, the growth of two strains of *Oscillatoria agardhii* was the lowest (~0.01 mg $mL^{-1}$ dw), while toxin production was high at 1.5–2 µg $L^{-1}$ vs. a toxin concentration of 3–5 µg $L^{-1}$ at ~0.2 mg $mL^{-1}$ dw [58].

In this study, all measured environmental variables in the water samples of the Lichk, Norashen and Lchashen bays were far lower than the "average" water quality index (Table 1), approaching the "good" or even "excellent" water quality criteria adopted in Armenia. However, most of the genotoxic responses of the *Tradescantia* inflorescences exposed to the water samples were significantly greater than that found in the control (Figure 6). While it is difficult to predict a cause-and-effect relationship between blooms and environmental conditions [4], the proliferation of algae and cyanobacteria and the presence of toxins in the water may be one of the factors causing such genotoxic responses. Moreover, the effect of unmeasured environmental agents cannot be excluded. The results of the integrated PCA analysis on the available biological and ecotoxicological data showed that all genotoxicity indicators (PC, CC, SH and MN) were grouped with the total numbers of either Cyanophyta or Chlorophyta measured in the bays' water (Figure 7). A more detailed PCA analysis with individual genera of Cyanophyta and Chlorophyta grouped the frequency of SH occurrences with the abundance of *Anabaena* (*Dolichospermum/Anabaena flos-aquae)* and *Oscillatoria* sp. at all sites during the spring and summer (Figure 8). The SH frequencies were similar across the sites and significantly exceeded that of the control (Figure 6). The numbers of *Oscillatoria* sp. were higher in the spring and summer (at Norashen), while *Anabaena* sp. bloomed in the summer (at Lichk) (Figure 5). Both species produce neurotoxins (e.g., anatoxins), while *Anabaena* also produces dissolved microcystins, cyanobacterial secondary metabolites [58–60]. While both microcystin and anatoxin-a can be degraded to nontoxic degradation products [61–63], microcystins can persist for weeks [64,65]. Even if a bloom has collapsed, microcystins can still exist for several months [66] and be transported over long distances [67]. The persistence of microcystin variants leads to their accumulation in surface waters [66]. Hence, elevated cyanotoxin concentrations could be expected at the Lichk and Norashen bays to trigger the occurrence of SH in the *Tradescantia*.

Although, with lower factor loadings, the occurrences of PC were grouped with the abundance of *Phormidium* sp. at Norashen Bay in the autumn and the MN occurrences were grouped with *Oscillatoria* sp. at Norashen in the summer (Figure 8). In both seasons, these genotoxicity indicators were the greatest at Norashen compared to the other bays (Figure 5). Phormidium sp. dominated the phytoplankton community in the autumn at Norashen. *Oscillatoria* sp. development mainly occurred at Norashen during spring and summer. Both species are known for forming benthic mats, which later start to float in stagnant waters. Moreover, these genera contain toxin-producing taxa [68]. An availability of mats likely containing toxins could have contributed to the greatest values of PC and MN at Norashen compared to the other sites. Such attributions may be further supported by the fact that the genotoxicity indicators of different sensitivities (highly sensitive MN and less sensitive PC) were the greatest at Norashen Bay. Gerashkin et al. [69] mentioned that the Trad-MN test is more sensitive in comparison with the Trad-SHM test due to the appearance of micronuclei at any damaged site of any of the 12 chromosomes of *Tradescantia*, in contrast to PC, which happens only at one locus in one chromosome in the stamen hair cell in the plant.

The PCA grouped the frequency of CC occurrences with the abundance of *Aphanothece* sp. in all autumn samples (Figure 8). Similar to *Phormidium* sp., the *Aphanothece* sp. also forms benthic mats and also includes toxin-producing taxa [68]. Higher occurrences of CC in the *Tradescantia* stamen hair may reasonably be attributed to the accumulation of cyanotoxins in the water bodies by the autumn.

A detailed PCA with Chlorophyta sp. grouped the SH frequencies with the numbers of green alga *Botryococcus* sp. at Lichk bay in the spring and summer (Figure 9). The *Botryococcus* sp. was represented by *B. braunii* in the studied bays. Their numbers were the greatest at Lichk compared to other sites. *B. braunii* releases free fatty acids (when cells disintegrate, e.g., at a later stage of blooming), which can be toxic to aquatic organisms, although the relationship between the production and disintegration of the toxin remains to be understood [70]. The allelopathic effect of the free fatty acids of *B. braunii* on phytoplankton was attributed to the suppressed diversity of phytoplankton in Lake Liyu (Taiwan) in 1998 and 1999 at *B. braunii* numbers from ~12,000 to 24,000 cells L$^{-1}$, correspondingly. Moreover, testing with the extracts of different fatty acids from the alga showed toxicity to various zooplankton species. Finally, fish deaths, mainly *Tilapia* sp., were also associated with its bloom [70]. In Lichk Bay, the numbers of *B. braunii* amounted to 48,000 and 24,000 cells L$^{-1}$ in the spring and summer, respectively, being similar to or even exceeding those reported for Lake Liyu. One may assume that this alga can be toxic to *Tradescantia* inflorescences as well, in particular, affecting the growth of normal stamen hair and generating SH.

The PCA indicated a relatively weak relationship between the Bacillariophyta abundance and SH at Lichk Bay in the spring (Figure 7), although the bay's water was enriched in phosphate (>0.4 mg L$^{-1}$), and diatoms proliferated (>1.3 mln cells L$^{-1}$) (Figures 3 and 4). The diatom genera *Amphora*, *Pseudo-nitzschia* and *Nitzschia* include strains that produce domoic acid, a neurotoxin. Within cells, its domoic acid content is at a pico scale [71,72]; however, it can accumulate within the food web [72–74]. In the study bays, the *Amphora* genus was represented by *A. ovalis*, which was abundant in the spring at Lichk Bay. The *Nitzschia* genus was represented by several species, *N. dissipata, N. acicularis, N. angustata, N. palea* and *N. linearis*, which were also highly abundant at Lichk in the spring (Figure S2). However, we could not find data on the production of domoic acid or other toxins by these species. Although the presence of toxin-producing diatoms at Lichk cannot be excluded, and thus their potential effect on the occurrences of SH, the results of the PCA did not demonstrate a strong relationship between the diatom abundance and genotoxic responses of *Tradescantia*.

Suitability of *Tradescantia*-based test systems for the evaluation of toxic potential of HABs

While the reported HAB event occurred at Lichk Bay in the summer of 2020 (*Dolichospermum*/Anabaena flos-aquae reaching nearly 2 mln cells L$^{-1}$), strong relationships between the proliferation of green algae and cyanobacteria and the genotoxic responses of *Tradescantia* (clone 02) inflorescences were found by PCA for other bays and seasons as well. Both phyla (green algae and cyanobacteria) studied in this work included known toxin-producing taxa. Green alga *B. braunii* could adversely affect the production of short stamen hair (SH) in the plant. From cyanobacteria species, *Anabaena* and *Oscillatoria* sp. could also be the cause for the suppression of the growth of stamen hair (SH), while *Phormidium* sp. could trigger pink cell occurrence in the stamen hair (PC). *Aphanothece* sp. could be responsible for colorless cell occurrence (CC), and *Oscillatoria* sp. could be responsible for the occurrence of chromosomal damage in the pollen mother cells (MN) in inflorescences. Diatom proliferation in the spring was unlikely to trigger a genotoxic response in *Tradescantia*. Thus, *Tradescantia* inflorescences demonstrated both genotoxic (point mutations) and clastogenic (disturbances in the microsporogenesis) responses when exposed to natural water with a high abundance of toxin-producing algal/cyanobacterial taxa, in which the concentrations of other contaminants were below regulatory norms. The relationships between the genotoxic and clastogenic responses and individual toxin-producing genera were also demonstrated by PCA. These results suggest that the Trad-SHM and Trad-MN test systems can be used for the evaluation of the ecotoxicity of algal/cyanobacterial toxins in water, particularly during HABs.

## 5. Conclusions

The biological, ecotoxicological and integrated results of this study demonstrated the genotoxic (mutations in somatic cells in the stamen hair and stunted stamen hairs) and clastogenic (chromosomal aberrations during microsporogenesis in the pollen mother cells) responses in *Tradescantia* (clone 02) inflorescences. They were likely caused by the toxins produced by certain algal/cyanobacterial groups (green alga *B. braunii*, cyanobacteria *Anabaena*, *Oscillatoria*, *Phormidium* and *Aphanothece*) under the appropriate environmental conditions because the values of the regularly monitored chemical parameters in the studied waters corresponded to the good water quality classification. Hence, both standard tests, the Trad-SHM and Trad-MN, are suitable for the evaluation of the toxic potential of algal proliferation and HABs.

**Supplementary Materials:** The following supporting information can be downloaded at: https://www.mdpi.com/article/10.3390/w15132500/s1, Figure S1: Mean air temperature in Lichk, Norashen and Lchashen bays of Lake Sevan (Armenia) from May to October 2020; Figure S2: Total numbers of Bacillariophyta (diatom) genera/species in Lichk, Norashen and Lchashen bays of Lake Sevan (Armenia) in 2020 by season.

**Author Contributions:** Conceptualization, B.G. and R.A. (Rouben Aroutiounian); methodology, R.A. (Rima Avalyan), A.A., E.A. and L.H.; formal analysis, A.K.; investigation, R.A. (Rima Avalyan), A.A., E.A. and L.H.; resources, B.G.; writing—original draft preparation, A.K.; writing—review and editing, B.G. and R.A. (Rouben Aroutiounian); supervision, B.G. and R.A. (Rouben Aroutiounian); project administration, B.G.; funding acquisition, B.G. All authors have read and agreed to the published version of the manuscript.

**Funding:** This research was funded by the Science Committee of Armenia, grants 21SC-BRFFR-1F028 and 1-15/20TB. AK was supported by the institutional grant, Arengufond_AK.

**Data Availability Statement:** Data are available from the corresponding author on request.

**Conflicts of Interest:** The authors declare no conflict of interest.

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
