# Peer review of "Tradescantia-Based Test Systems Can Be Used for the Evaluation of the Toxic Potential of Harmful Algal Blooms"

_water, doi:10.3390/w15132500_

Round 1

Reviewer 1 Report

The manuscript describes for the first time the analysis of a potential toxic algal bloom in 2020 using two different Tradescantia-based test systems.  In general, the manuscript is interesting but the lack of a complete and systematic analysis of the causes of the blooms decreases the validity of the study.  The authors apparently do not fully understand basic algal physiology very well as evidenced by the limited number of environmental nutrient variables measured (only ammonium and phosphate) and neglected nitrate and organic forms that are normally utilized. 

The general writing style is good and easily understandable but there are several minor spelling and punctuation errors.

Specific comments:

line 46 - replace "enter" with "enters"

line 107 - change to read "an algal bloom"

line 111 - add comma to read "warming, HAB"

lines 145 and 346 - nitrate is a well known nutrient which should have been measured

line 397 - Armenia water quality standards are obviously much higher than meaningful nutrient concentrations utilized by algal species.

line 403 - "unmeasured environmental agents" is the basis of many comments in this review.

line 462 - change to read "Within cells its domoic acid content"

line 483 - correct "al7so" to read "also"

References 3, 22, 57 and 59 need second line to be right-justified

In general the English in the manuscript is good with only a few small corrections necessary.  The sampling design of variables measured in the lakes should have been more thoroughly considered.

Author Response

We thank the reviewer for her time and comments. Our responses are in the attached file.

Reviewer 2 Report

See comments file above.

Author Response

(The authors gave the same response as above.)

Reviewer 3 Report

Khosrovyan et al. evaluated the use of Tradescantia for the evaluation of the toxic potential of harmful algal blooms, concluding that Tradescantia-based systems (Trad-SHM and Trad-MN) may indeed be used for such purpose. The article is relevant, especially considering the need to advance the use of non-animal-based tests in toxicology. However, before publication, some aspects of the manuscript must be improved, as detailed below:

- Introduction, first paragraph: Please describe in more detail the detrimental impacts of algal blooms on non-human animals and ecosystems.

- Introduction, lines 53-54: Cite this link as a reference (the link must be cited as a reference in the reference list).

- Figure 1: Correct “Armeni” to “Armenia”. Also, a photograph of Lake Sevan's landscape could be added to Figure 1, as a new panel.

- Introduction, lines 105-106: Cite this link as a reference (the link must be cited as a reference in the reference list).

- Introduction, last paragraph: Give a more detailed description of the Tradescantia use in ecotoxicology.

- Introduction, general comment: Tradescantia must be cited in italics. Please correct this throughout the manuscript.

- Statistical analysis: All packages used in R must be cited (for analysis, graphs, etc). Also, R must be cited as a reference and included in the reference list. Moreover, if other software was used to plot some graphs (e.g., Excel), it must also be cited.

- 3.1 Chemical analysis, lines 212-213: Cite this link as a reference (the link must be cited as a reference in the reference list).

- Figure 5, panel A: The columns/bars are overlapping. Please fix this to make it consistent with panel B.

- Results, general comment: A representative figure of Trad-SHM and Trad-MN systems would be a nice addition to the results section.

- Discussion, lines 342-343: Cite this link as a reference (the link must be cited as a reference in the reference list).

- Discussion: More hypotheses for the 2020 algal bloom (anthropogenic triggers?) could be added to the discussion section.

- Discussion, line 391: Cite M. aeruginosa in italics (please revise all scientific names throughout the manuscript; same issue for lines 444, 453 and 454, for example).

- Discussion, Suitability of Tradescantia-based test systems for the evaluation of toxic potential of HABs: This discussion subsection must be more robust. It is not clear how or why Tradescantia-based test systems are proper for the ecotoxicological evaluation performed by the authors. This section must show clearly to the reader that Tradescantia-based tests are useful for this and other similar studies.

- Conclusions, line 494: “Effects of” what specifically?

- Figure S2: The columns/bars are overlapping. Please fix this.

English is understandable.

Author Response

(The authors gave the same response as above.)

Round 2

Reviewer 3 Report

The authors improved the article sufficiently.

English is understandable.